# Adaptive evolution of nontransitive fitness in yeast

**Sean W Buskirk†, Alecia B Rokes, Gregory I Lang***

Department of Biological Sciences, Lehigh University, Bethlehem, United States

**Abstract** A common misconception is that evolution is a linear 'march of progress', where each organism along a line of descent is more fit than all those that came before it. Rejecting this misconception implies that evolution is nontransitive: a series of adaptive events will, on occasion, produce organisms that are less fit compared to a distant ancestor. Here we identify a nontransitive evolutionary sequence in a 1000-generation yeast evolution experiment. We show that nontransitivity arises due to adaptation in the yeast nuclear genome combined with the stepwise deterioration of an intracellular virus, which provides an advantage over viral competitors within host cells. Extending our analysis, we find that nearly half of our ~140 populations experience multilevel selection, fixing adaptive mutations in both the nuclear and viral genomes. Our results provide a mechanistic case-study for the adaptive evolution of nontransitivity due to multilevel selection in a 1000-generation host/virus evolution experiment.

**\*For correspondence:**
glang@lehigh.edu

**Present address:** †Department of Biology, West Chester University, West Chester, United States

**Competing interests:** The authors declare that no competing interests exist.

## Introduction

Adaptive evolution is a process in which selective events result in the replacement of less-fit genotypes with a more fit ones. Intuitively, a series of selective events, each improving fitness relative to the immediate predecessor, should translate into a cumulative increase in fitness relative to the ancestral state. However, whether or not this is borne out over long evolutionary time scales has long been the subject of debate (*Ruse, 1993*; *Dawkins, 1997*; *Gould, 1997*; *Shanahan, 2000*). The failure to identify broad patterns of progress over evolutionary time scales—despite clear evidence of selection acting over successive short time intervals—is what *Gould, 1985* referred to as 'the paradox of the first tier'. This paradox implies that evolution exhibits nontransitivity, a property that is best illustrated by the Penrose staircase and the Rock–Paper–Scissors game. The Penrose staircase is a visual illusion of ascending sets of stairs that form a continuous loop such that—although each step appears higher than the last—no upward movement is realized. In the Rock–Paper–Scissors game each two-way interaction has a clear winner (paper beats rock, scissors beats paper, and rock beats scissors), yet due to the nontransitivity of these two-way interactions, no clear hierarchy exists among the three.

In ecology, nontransitive interactions among extant species are well documented as contributors to biological diversity and community structure (*Kerr et al., 2002*; *Károlyi et al., 2005*; *Laird and Schamp, 2006*; *Reichenbach et al., 2007*; *Menezes et al., 2019*) and arise by way of resource (*Sinervo and Lively, 1996*; *Precoda et al., 2017*) or interference competition (*Kirkup and Riley, 2004*). First put forward in the 1970s (*Gilpin, 1975*; *Jackson and Buss, 1975*; *May and Leonard, 1975*; *Petraitis, 1979*), the importance of nontransitivity in ecology has garnered extensive theoretical and experimental consideration over the last half century (e.g. *Sinervo and Lively, 1996*; *Kerr et al., 2002*; *Allesina and Levine, 2011*; *Rojas-Echenique and Allesina, 2011*; *Soliveres et al., 2015*; *Liao et al., 2019*).

What is unknown is whether nontransitive interactions arise for direct descendants along a line of genealogical succession. This is the crux of Gould's paradox and has broad implications for our understanding of evolutionary processes. For instance, if an evolved genotype is found to be less fit

**eLife digest** It is widely accepted in biology that all life on Earth gradually evolved over billions of years from a single ancestor. Yet, there is still much about this process that is not fully understood. Evolution is often thought of as progressing in a linear fashion, with each new generation being better adapted to its environment than the last. But it has been proposed that evolution is also nontransitive: this means even if each generation is 'fitter' than its immediate predecessor, these series of adaptive changes will occasionally result in organisms that are less fit than their distant ancestors.

Laboratory experiments of evolution are a good way to test evolutionary theories because they allow researchers to create scenarios that are impossible to observe in natural populations, such as an organism competing against its extinct ancestors. Buskirk et al. set up such an experiment using yeast to determine whether nontransitive effects can be observed in the direct descendants of an organism.

At the start of the experiment, the yeast cells were host to a non-infectious 'killer' virus that is common among yeast. Cells containing the virus produce a toxin that destroys other yeast that lack the virus. The populations of yeast were given a nutrient-rich broth in which to grow and subjected to a simple evolutionary pressure: to grow fast, which limits the amount of resources available.

As the yeast evolved, they gained beneficial genetic mutations that allowed them to outcompete their neighbors, and they passed these traits down to their descendants. Some of these mutations occurred not in the yeast genome, but in the genome of the killer virus, and this stopped the yeast infected with the virus from producing the killer toxin. Over time, other mutations resulted in the infected yeast no longer being immune to the toxin. Thus, when Buskirk et al. pitted these yeast against their distant ancestors, the new generation were destroyed by the toxins the older generation produced.

These findings provide the first experimental evidence for nontransitivity along a line of descent. The results have broad implications for our understanding of how evolution works, casting doubts over the idea that evolution always involves a direct progression towards new, improved traits.

in comparison to a distant ancestor, the adaptive landscape upon which the population is evolving may not contain true fitness maxima (*Barrick and Lenski, 2013*; *Van den Bergh et al., 2018*) and, more broadly, directionality and progress may be illusory (*Gould, 1996*). Testing the hypothesis that nontransitive interactions arise along lines of genealogical descent, however, is not possible in natural populations because it requires our ability to directly compete an organism against its immediate predecessor as well as against its extinct genealogical ancestors. Fortunately, laboratory experimental evolution, in which populations are preserved as a 'frozen fossil record', affords us with the unique opportunity to test for nontransitivity along a genealogical lineage by directly competing a given genotype against the extant as well as the extinct.

An early study of laboratory evolution of yeast in glucose-limited chemostats appeared to demonstrate that nontransitive interactions arise along a line of genealogical descent (*Paquin and Adams, 1983*). However, the specific events that led to nontransitivity in this case are unknown, and it is likely the case that the authors were measuring interactions between contemporaneous lineages in a population, rather than individuals along a direct line of genealogical descent, as they report (see Discussion). Indeed, adaptive diversification is common in experimental evolution due to spatial structuring (*Rainey and Travisano, 1998*; *Frenkel et al., 2015*) and metabolic diversification (*Paquin and Adams, 1983*; *Helling et al., 1987*; *Turner et al., 1996*; *Spencer et al., 2008*; *Kinnersley et al., 2014*), and is typically maintained by negative frequency-dependent selection, in which rare genotypes are favored. Collectively this work reinforces theory and observational evidence on the power of ecological nontransitivity as a driver and maintainer of diversity but is silent as to whether genealogical succession can also be nontransitive.

Here we determine the sequence of events leading to the evolution of nontransitivity in a single yeast population during a 1000-generation evolution experiment. We show that nontransitivity arises through multilevel selection involving both the yeast nuclear genome and the population of a vertically transmitted virus. Many fungi, including the yeast *Saccharomyces cerevisiae*, are host to

non-infectious, double-stranded RNA 'killer' viruses (*Wickner, 1976*; *Schmitt and Breinig, 2002*; *Schmitt and Breinig, 2006*; *Rowley, 2017*). Killer viruses produce a toxin that kills non-killer containing yeasts. The K1 toxin gene contains four subunits (δ, α, γ, and β), which are post-translationally processed and glycosylated to produce an active two-subunit (α, β) secreted toxin (*Bostian et al., 1983*). Immunity to the toxin is conferred by the pre-processed version of the toxin, thus requiring cells to maintain the virus for protection. We show that nontransitivity arises due to multilevel selection: adaptation in the yeast nuclear genome and the simultaneous stepwise deterioration of the killer virus. By expanding our study of host-virus genome evolution to over 100 additional yeast populations, we find that multilevel selection, and thus the potential for the evolution of nontransitive interactions, is a common occurrence given the conditions of our evolution experiment.

## Results

### Evolution of nontransitivity along a line of genealogical descent

Previously we evolved ~600 haploid populations of yeast asexually for 1000 generations in rich glucose medium (*Lang et al., 2011*). We characterized extensively the nuclear basis of adaptation for a subset of these populations through whole-genome whole-population time-course sequencing (*Lang et al., 2013*) and/or fitness quantification of individual mutations (*Buskirk et al., 2017*).

For one population (BYS1-D08) we were surprised to observe that a 1000-generation clone lost in direct competition with a fluorescently labeled version of the ancestor. To test the hypothesis that a nontransitive interaction arose during the adaptive evolution of this population, we isolated individual clones from three time points: Generation 0 (Early), Generation 335 (Intermediate), and Generation 1000 (Late) (*Figure 1A*). These time points were chosen, in part, to coincide with the completion of selective sweeps in the population (*Lang et al., 2013*). The Intermediate clone was isolated following a selective sweep that fixes three nuclear mutations including a beneficial mutation in *YUR1*. The Late clone was isolated following three more selective sweeps that fix an additional 10 nuclear mutations including a beneficial mutation in *STE4*.

We performed pairwise competition experiments between the Early, Intermediate, and Late clones at multiple starting frequencies. We find that the Intermediate clone is 3.8% more fit relative to the Early clone and that the Late clone is 1.2% more fit relative to the Intermediate clone (*Figure 1B*, left panel). The *yur1* mutation in the Intermediate clone and the *ste4* mutation in the late clone were previously estimated to provide a $4.6 \pm 0.5\%$ and $2.6 \pm 0.4\%$ fitness advantage, respectively (*Buskirk et al., 2017*), consistent with the fitness differences between the Intermediate and Early clones and the Late and Intermediate clones. The expectation, assuming additivity, is that the Late clone will be more fit than the Early clone, by roughly 5.0%. Surprisingly, we find that the Late clone is less fit than expected, to the extent that it often loses in pairwise competition with the Early clone (*Figure 1B*, left panel). Furthermore, the interaction between the Early and Late clones exhibits positive frequency-dependent selection, thus creating a bi-stable system where the fitness disadvantage of the Late clone can be overcome if it starts above a certain frequency relative to the Early clone (*Figure 1—figure supplement 1*).

### Evolution of nontransitivity is associated with changes to the killer virus

Positive frequency-dependent selection is rare in experimental evolution and can only arise through a few known mechanisms. It has been observed previously in yeast that harbor killer viruses (*Greig and Travisano, 2008*), which are dsRNA viruses that encode toxin/immunity systems. Using a well-described halo assay (*Woods and Bevan, 1968*), we find that the ancestral strain of our evolved populations exhibits the phenotype expected of yeast that harbor the killer virus: it inhibits growth of a nearby sensitive strain and resists killing by a known killer strain (*Figure 1—figure supplement 2*).

We asked if the observed nontransitivity in the BYS1-D08 lineage could be explained by evolution of the killer phenotype. Phenotyping of the isolated clones revealed that the Intermediate clone no longer exhibits killing ability (K$^-$I$^+$) and that the Late clone possesses neither killing ability nor immunity (K$^-$I$^-$, *Figure 1A*, *Figure 1—figure supplement 3*). Killer toxin has been shown to impart frequency-dependent selection in structured environments due to a high local concentration of secreted toxin (*Greig and Travisano, 2008*). We hypothesized that a stepwise loss of killing ability

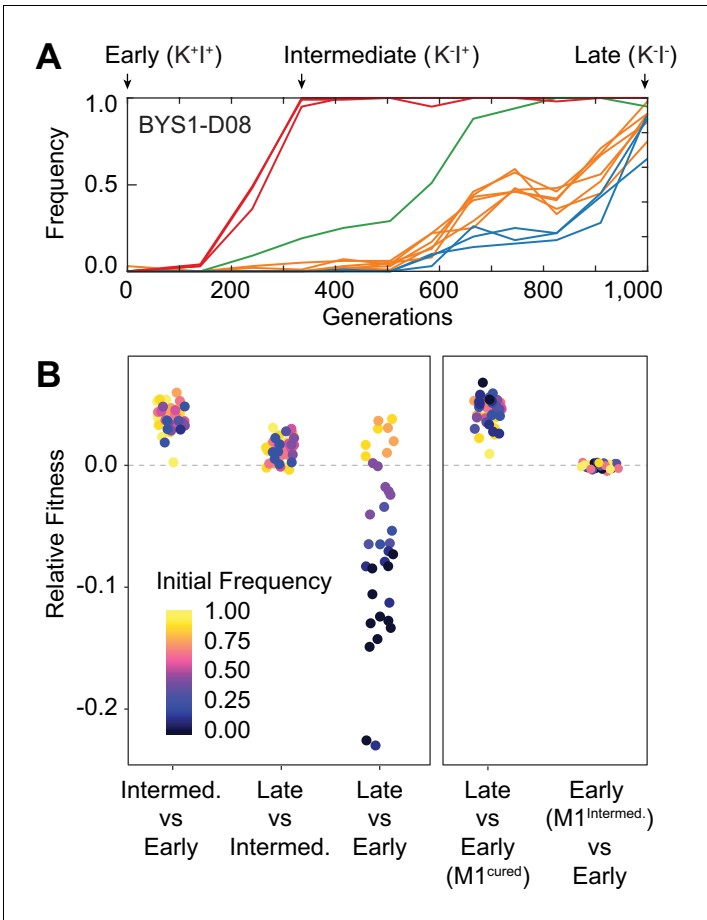

**Figure 1.** Nontransitivity and positive frequency dependence arise along an evolutionary lineage. (A) Sequence evolution (from *Lang et al., 2013*) shows that population BYS1-D08 underwent four clonal replacements over 1000 generations. Mutations in the population that went extinct are not shown. The four selective sweeps are color-coded: red, mutations in *yur1*, *rxt2*, and an intergenic mutation; green, a single intergenic mutation; orange, mutations in *mpt5*, *gcn2*, *iml2*, *ste4*, *mud1*, and an intergenic mutation; blue, three intergenic mutations. The Intermediate clone isolated at Gen. 335 does not produce, but is resistant to, the killer toxin (K⁻I⁺). The Late clone, isolated at Generation 1000 does not produce, and is sensitive to, the killer toxin (K⁻I⁻). (B) Competition experiments demonstrate nontransitivity and positive frequency-dependent selection. Left: Relative fitness of Early (Gen. 0), Intermediate (Gen. 335), and Late (Gen. 1000) clones. Right: Relative fitness of the Early clone without ancestral virus or with the viral variant from the Intermediate clone. Fitness and starting frequency correspond to the later clone relative to the earlier clone during pairwise competitions.

The online version of this article includes the following figure supplement(s) for figure 1:

**Figure supplement 1.** Positive frequency-dependent interaction along an evolutionary lineage.

**Figure supplement 2.** Visualization of killer phenotype by halo assay.

**Figure supplement 3.** Stepwise deterioration of killer phenotype in evolved clones.

followed by loss of immunity, along with the acquisition of beneficial *yur1* and *ste4* nuclear mutations, were responsible for the frequency-dependent and nontransitive interaction between Early and Late clones.

To determine if killer toxin production by the Early clone is necessary for it to outcompete the toxin-susceptible Late clone, we repeated the competition between the Early and Late clones using a virus-cured version of the Early clone. We find that removing the virus from the Early clone abolishes the frequency-dependent fitness advantage of the Early clone; the Late clone is 4.3% more fit than the cured Early clone at all frequencies (*Figure 1B*, right panel) due to the presence of adaptive mutations in the nuclear genome of the Late clone. Therefore the presence of killer virus in the Early

clone and the subsequent loss of killer virus-associated phenotypes in the Late clone were necessary for the evolution of frequency dependence and nontransitivity.

To determine if viral evolution alone is sufficient to account for the observed fitness gains in non-transitive interactions, we focused on the first step in the evolutionary sequence: the transition from the Early clone to Intermediate clone. We transferred the killer virus from the Intermediate clone to the cured Early clone and assayed fitness relative to the Early clone. Because the virus from the Intermediate clone no longer produces toxin, we suspected that it may provide a fitness benefit to the host. However, we find that the evolved killer virus from the Intermediate clone confers no significant effect on host fitness compared to the killer virus from the Early clone (*Figure 1B*, right panel). This shows that the fitness benefit of the Intermediate clone relative to the Early clone is due to adaptation in the nuclear genome. Taken together these experiments show that the sequence of events leading to the evolution of nontransitivity involves changes to both the host and viral genomes.

## Changes to killer-associated phenotypes are common under our experimental conditions

To determine the extent of killer phenotype evolution across all populations, we assayed the killer phenotype of 142 populations that were founded by a single ancestor and propagated at the same bottleneck size as BYS1-D08 (*Lang et al., 2011*). We find that approximately half of all populations exhibit a loss or weakening of killing ability by Generation 1000, with ~10% of populations exhibiting neither killing ability nor immunity (*Figure 2*). Of note, we did not observe loss of immunity without loss of killing ability, an increase in killing ability or immunity, or reappearance of killing ability or immunity once it was lost from a population (*Figure 2—figure supplement 1*), apart from the noise associated with scoring of population-level phenotypes. Several populations (i.e. BYS2-B09 and BYS2-B12) lost both killing ability and immunity simultaneously, suggesting that a single event can cause the loss of both the killer phenotypes.

Mutations in nuclear genes can affect killer-associated phenotypes. The primary receptors of the K1 killer toxin are β-glucans in the yeast cell wall (*Pieczynska et al., 2013*). We observe a statistical

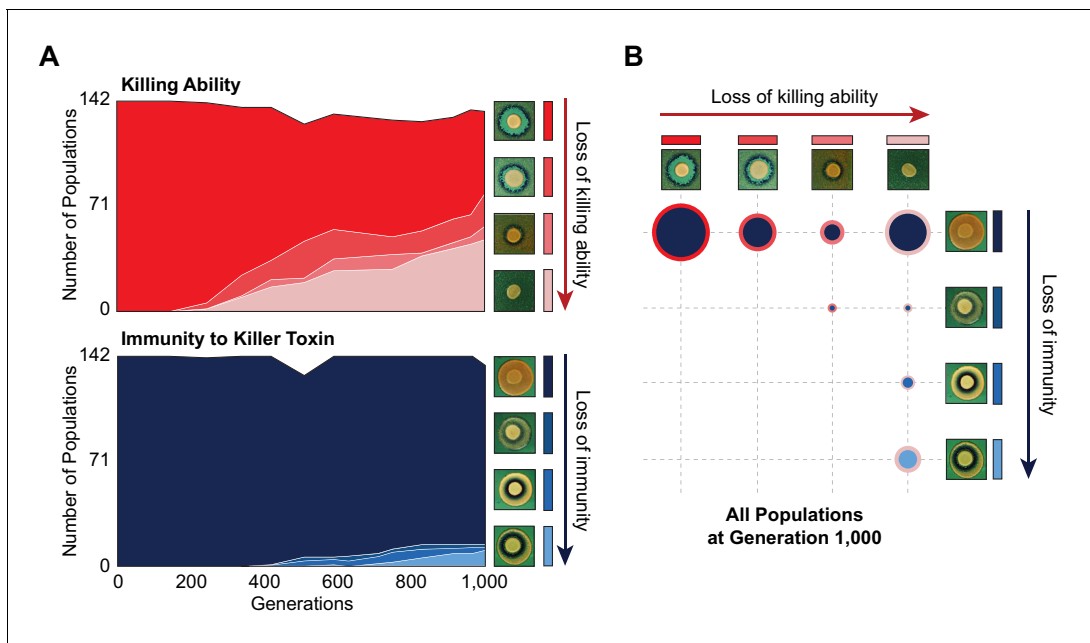

**Figure 2.** Changes in killer-associated phenotypes in the 142 populations that were founded by a single ancestor and propagated at the same bottleneck size as BYS1-D08 (*Lang et al., 2011*). (**A**) Loss of killing ability (top) and immunity (bottom) from evolving yeast populations over time. Killer phenotypes were monitored by halo assay (examples shown on right). (**B**) Breakdown of killer phenotypes for all populations at Generation 1000. Data point size corresponds to number of populations. Border and fill color indicate killing ability and immunity phenotypes, respectively, as in panel A. The online version of this article includes the following figure supplement(s) for figure 2:

**Figure supplement 1.** Killer phenotypes of the 17 populations that develop sensitivity to the K1 toxin.

enrichment of mutations in genes involved in β-glucan biosynthesis (sixfold Gene Ontology [GO] Biological Process enrichment, p<0.0001). Furthermore, of the 714 protein-coding mutations dispersed across 548 genes, 40 occur within 11 of the 36 genes (identified by *Pagé et al., 2003*) that, when deleted, confer a high level of resistance to the K1 toxin ($\chi^2$ = 18.4, *df* = 1, p=1.8 × 10$^{-5}$). Nevertheless, the presence of mutations in nuclear genes that have been associated with high levels of resistance is not sufficient to account for the loss of killing ability ($\chi^2$ = 1.037, *df* = 1, p=0.309) or immunity ($\chi^2$ = 0.103, *df* = 1, p=0.748).

## Standing genetic variation and de novo mutations drive phenotypic change

We sequenced viral genomes from our ancestral strain and a subset of yeast populations (n = 67) at Generation 1000 (*Figure 3*). We find that our ancestral strain, which was derived from the common lab strain W303-1a, contains the M1-type killer virus (encoding the K1-type killer toxin) with only minor differences from previously sequenced strains (*Figure 3—figure supplement 1*). Our ancestral strain also possesses the L-A helper virus, which supplies the RNA-dependent RNA polymerase and capsid protein necessary for the killer virus, a satellite virus, to complete its life cycle (*Ribas and Wickner, 1992*). We sequenced viral genomes from 57 populations that changed killer phenotype and 10 control populations that retained the ancestral killer phenotypes. Viral genomes isolated from populations that lost killing ability possess 1–3 mutations in the M1 coding sequence – most being missense variants (*Figure 3A*). In contrast, only a single mutation, synonymous nonetheless, was detected in M1 across the 10 control populations that retained the killer phenotype. The correlation between the presence of mutations in the viral genome and the loss of killing ability ($\chi^2$ = 59.3, *df* = 1, p=1.4 × 10$^{-13}$) is strong evidence that viral mutations are responsible for the changes in killer phenotypes. We estimate that by Generation 1000 half of all populations have fixed viral variants that alter killer phenotypes (for comparison, *IRA1*, the most common nuclear target, fixed in ~25% of populations over the same time period).

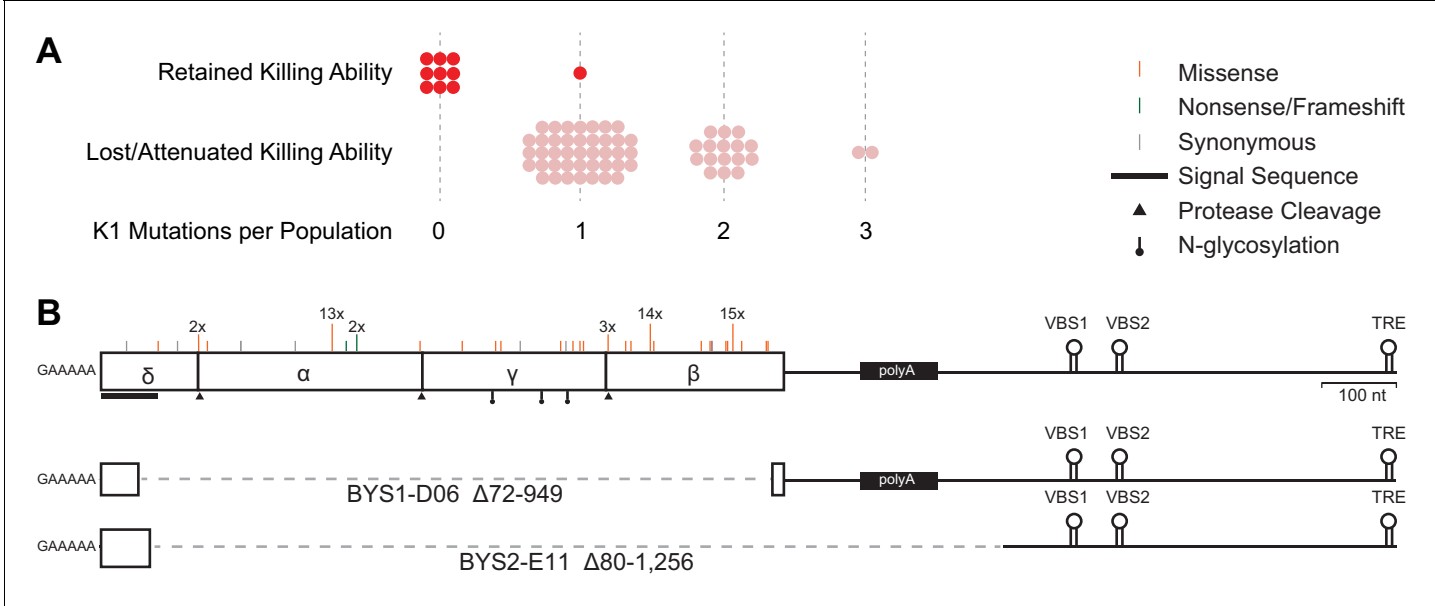

**Figure 3.** Loss of killer phenotype correlates with the presence of mutations in the K1 toxin gene. (A) Number of mutations in the K1 gene in yeast populations that retain or lose killing ability. Each data point represents a single yeast population. (B) Observed spectrum of point mutations across the K1 toxin in 67 evolved yeast populations. Mutations were detected in a single population unless otherwise noted. Large internal deletion variants from two yeast populations (BYS1-D06 and BYS2-E11). The deletions span the region indicated by the dashed gray line. VBS: viral binding site. TRE: terminal recognition element.

The online version of this article includes the following figure supplement(s) for figure 3:

**Figure supplement 1.** Sequence divergence of ancestral viruses.

Of the 57 populations that lost killing ability, 42 fixed one of three single nucleotide polymorphisms, resulting in amino acid substitutions D106G, D253N, and I292M and observed 13, 14, and 15 times, respectively (*Supplementary file 1*). Given their prevalence, these polymorphisms likely existed at low frequency in the shared ancestral culture (indeed, we can detect one of the common polymorphisms, D106G, in individual clones at the Early time point, indicating that this mutation was heteroplasmic in cells of the founding population). Killer phenotypes are consistent across populations that fixed a particular ancestral polymorphism (*Supplementary file 1*).

In addition to the three ancestral polymorphisms, we detect 34 putative de novo point mutations that arose during the evolution of individual populations (*Supplementary file 1*). Mutations are localized to the K1 coding sequence, scattered across the four encoded subunits, and skewed toward missense mutations relative to nonsense or frameshift (*Figure 3B*). Of the 78 identified mutations, 14 are predicted to fall at or near sites of protease cleavage or post-translational modification necessary for toxin maturation. Overall, however, the K1 coding sequence appears to be under balancing selection (dN/dS = 0.90), indicating that certain amino acid substitutions (e.g. those that eliminate immunity but retain killing ability) are not tolerated. In addition, substitutions are extremely biased toward transitions over transversions (*Supplementary file 2*, R = 6.4, $\chi^2$ = 44.2, *df* = 1, p<0.0001), a bias that is also present in other laboratory-derived M1 variants (R = 4.1) (*Suzuki et al., 2015*) and natural variation of the helper L-A virus (R = 3.0) (*Diamond et al., 1989*; *Icho and Wickner, 1989*). The transition:transversion bias appears specific to viral genomes as the ratio is much lower within evolved nuclear genomes (R = 0.8), especially in genes inferred to be under selection (R = 0.5), suggesting a mutational bias of the viral RNA-dependent RNA polymerase (*Lang et al., 2013*; *Fisher et al., 2018*; *Marad et al., 2018*).

Though point mutations are the most common form of evolved variation, we also detected two viral genomes in which large portions of the K1 ORF are deleted (*Figure 3B*). Despite the loss of the majority of the K1 coding sequence, the deletion mutants maintain cis signals for replication and packaging (*Ribas and Wickner, 1992*; *Ribas et al., 1994*). Notably, the two populations that possess these deletion mutants also possess full-length viral variants. The deletion mutants we observe are similar to the ScV-S defective interfering particles that have been shown to outcompete full-length virus presumably due to their decreased replication time (*Kane et al., 1979*; *Ridley and Wickner, 1983*; *Esteban and Wickner, 1988*).

## Host/virus co-evolutionary dynamics are complex and operate over multiple scales

To compare the dynamics of viral genome evolution, nuclear genome evolution, and phenotypic evolution we performed time-course sequencing of viral genomes from three yeast populations that lost killing ability and for which we have whole-population, whole-genome, and time-course sequencing data for the nuclear genome (*Lang et al., 2013*). As with the evolutionary dynamics of the host genome, the dynamics of viral genome evolution feature clonal interference (competition between mutant genotypes), genetic hitchhiking (an increase in frequency of an allele due to genetic linkage to a beneficial mutation), and sequential sweeps (*Figure 4*, *Figure 4—figure supplement 1*). Interestingly, viral sweeps often coincide with nuclear sweeps. Since the coinciding nuclear sweeps often contain known driver mutations, it is possible that the viral variants themselves are not driving adaptation but instead hitchhiking on the back of beneficial nuclear mutations. This is consistent with the observation that the introduction of the viral variant from the Intermediate clone did not affect the fitness of the Early clone (*Figure 1B*).

To determine if the loss of killer phenotype is caused solely by mutations in the killer virus, we transferred the ancestral virus (K⁺I⁺) and five evolved viral variants into the virus-cured Early clone via cytoduction (*Figure 5A*). The five viral variants were selected to span the range of evolved killer phenotypes: one exhibited weak killing ability and full immunity (KʷI⁺: D253N), three exhibited no killing ability and full immunity (K⁻I⁺: P47S, D106G, I292M), and one exhibited neither killing ability nor immunity (K⁻I⁻: −1 frameshift). Following cytoduction, we observed that the killer phenotype of each cytoductant matched the killer phenotype of the population of origin, which demonstrates that viral mutations are sufficient to explain changes in killer phenotypes (*Figure 5—figure supplement 1*).

To determine if any viral variants affect host fitness, we competed all five cytoductants against the killer-containing Early clone (K⁺I⁺) and the virus-cured Early clone (K⁻I⁻) in pairwise fashion. Frequency-dependent selection was observed in all cases in which one competitor exhibited killing

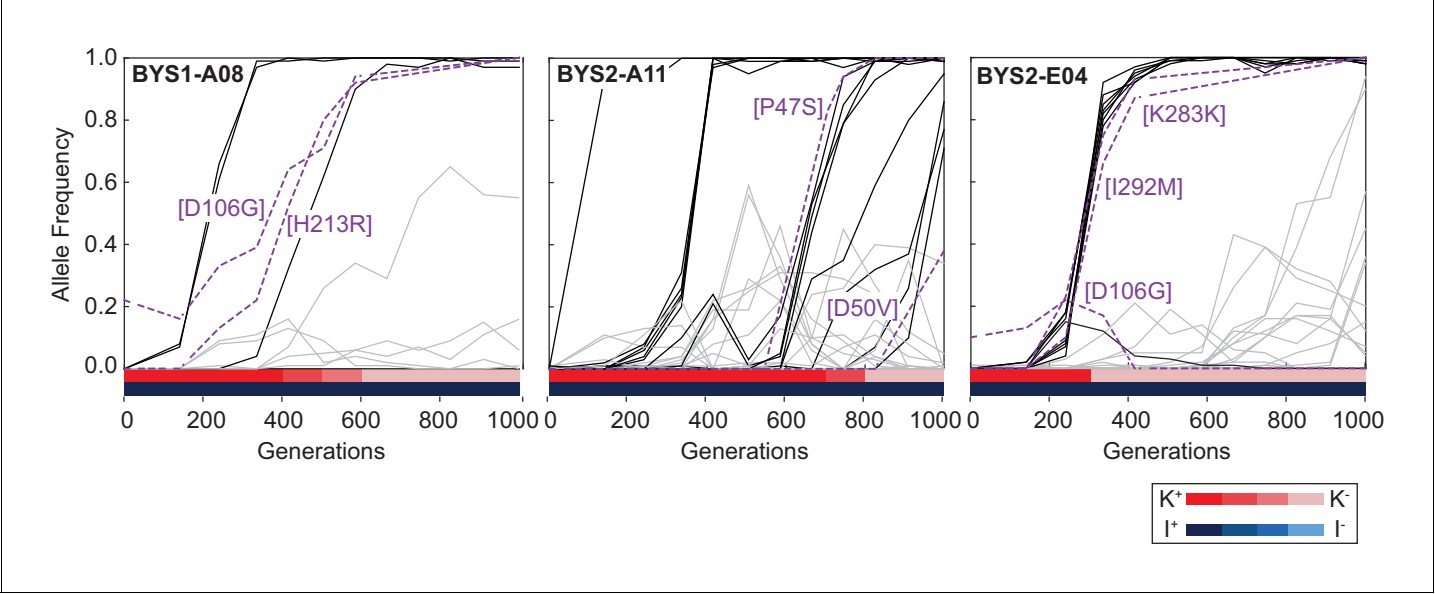

**Figure 4.** Viral dynamics mimic nuclear dynamics. Killer phenotype of evolved populations is indicated by color according to the key. Nuclear dynamics (reported previously in *Lang et al., 2013*) are represented as solid lines. Nuclear mutations that sweep before or during the loss of killing ability are indicated by black lines. All other mutations are indicated by gray lines. Viral mutations are indicated by purple dashed lines and labeled by amino acid change.

The online version of this article includes the following figure supplement(s) for figure 4:

**Figure supplement 1.** Evolutionary dynamics of nuclear genotypes and killer phenotypes over time.

ability and the other competitor lacked immunity (*Figure 5A*). For example, cytoductants containing either the ancestral virus or the weak-killing D253N variant exhibited a frequency dependent advantage over the virus-cured Early clone. However, in all competitions where the killer-associated phenotypes were compatible, host fitness was not impacted by the specific viral variant, or even by the presence of the virus itself. These data suggest that production of active toxin and maintenance of the virus have no detectable fitness costs to the host. These findings support previous theoretical and empirical studies (*Pieczynska et al., 2016*; *Pieczynska et al., 2017*) that claim that mycoviruses and their hosts have co-evolved to minimize cost.

## Success of evolved viral variants is due to an intracellular fitness advantage

Based on the lack of a measurable effect of viral mutations on host fitness when killing-mediated interactions are absent, we hypothesized that the evolved viral variants may have a selective advantage within the viral population of individual yeast cells. A within-cell advantage has been invoked to explain the invasion of internal deletion variants (e.g. ScV-S [*Kane et al., 1979*]) but has not been extended to point mutations. To test evolved viral variants for a within-cell fitness advantage, we generated a heteroplasmic diploid strain by mating the ancestor (with wild-type virus) with a haploid cytoductant containing either the I292M (K⁻I⁺) or −1 frameshift (K⁻I⁻) viral variant. The heteroplasmic diploids were propagated for seven single-cell bottlenecks every 48 hr to minimize among-cell selection. At each bottleneck, we assayed the yeast cells for killer phenotypes and we quantified the ratio of the intracellular viral variants by RT-PCR and sequencing. We find that killing ability was lost from all lines, suggesting that the evolved viral variants outcompeted the ancestral variant (*Figure 5B*). Sequencing confirmed that the derived viral variant fixed in most lines (*Figure 5C*). In some lines, however, the derived viral variant increased initially before decreasing late. Further investigation into one of these lines revealed that the decrease in frequency of the viral variant corresponded to the sweep of a de novo G131D variant (*Figure 5C*, inset). Viral variants therefore appear to constantly arise, and the evolutionary success of the observed variants results from their selective advantage over viral competitors within the context of an individual cell. We speculate that an intracellular

competition between newly arising viral variants also explains the loss of immunity from populations that previously lost killing ability (*Figure 2—figure supplement 1*), given the relaxed selection for the maintenance of functional immunity in those populations.

## Discussion

We examined phenotypic and sequence co-evolution of an intracellular double-stranded RNA virus and the host nuclear genome over the course of 1000 generations of experimental evolution. We

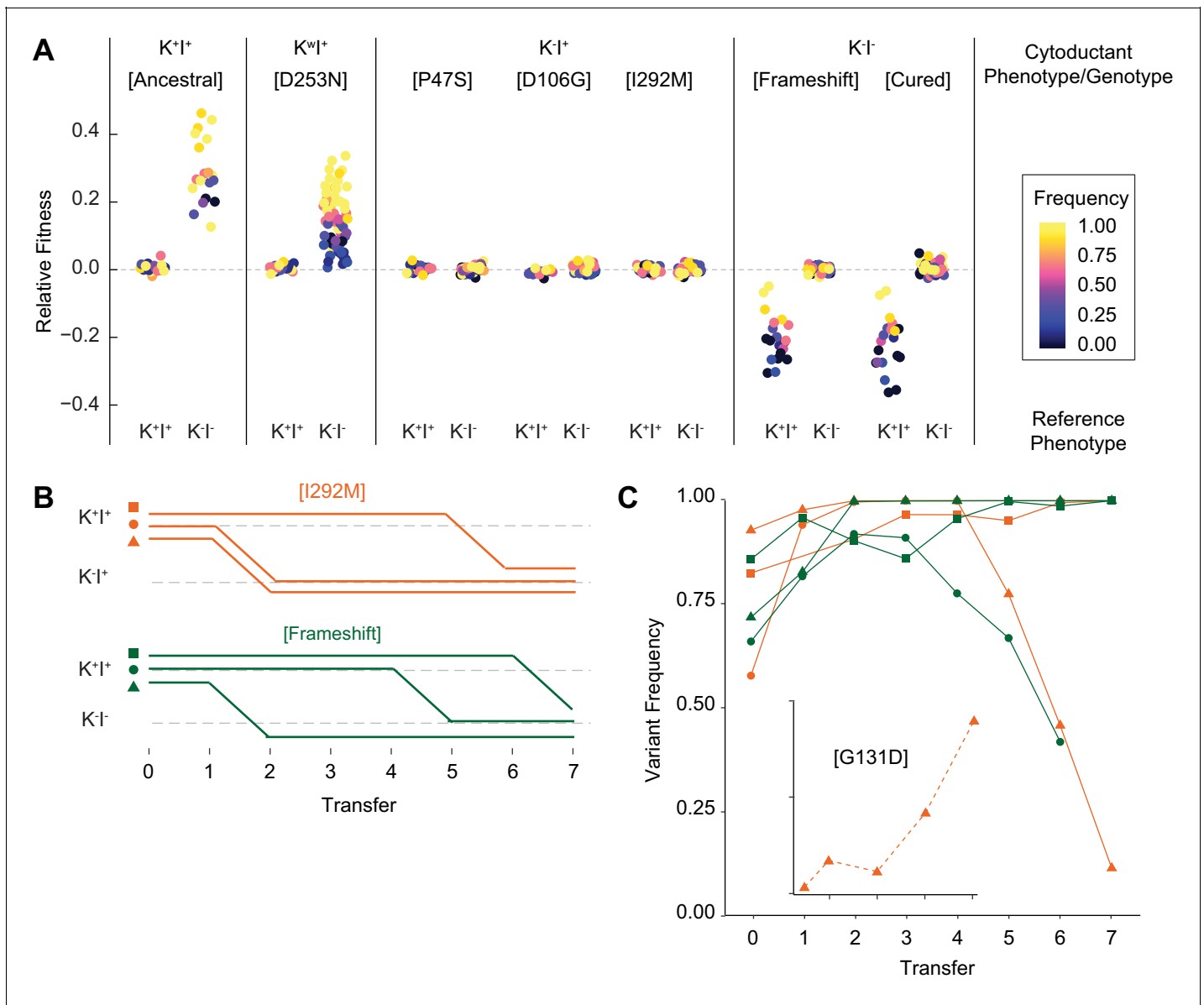

**Figure 5.** Viral evolution is driven by selection for an intracellular competitive advantage. (**A**) Relative fitness of viral variants in pairwise competition with the ancestor (K⁺I⁺) and virus-cured ancestor (K⁻I⁻). Killer phenotype and identity of viral variant labeled above (Kʷ indicates weak killing ability). Killer phenotype of the ancestral competitor labeled below. Starting frequency indicated by color. (**B**) Change to killer phenotype during intracellular competitions between viral variants (by color) and ancestral virus. Replicate lines indicated by symbol. (**C**) Variant frequency during intracellular competitions. Colors and symbols consistent with panel B. Inset: frequency of the de novo G131D viral variant.

The online version of this article includes the following figure supplement(s) for figure 5:

**Figure supplement 1.** Cytoductants exhibit the same killer phenotype as the population of origin.
**Figure supplement 2.** Consensus between Sanger and Illumina sequencing in reporting mutation frequency.

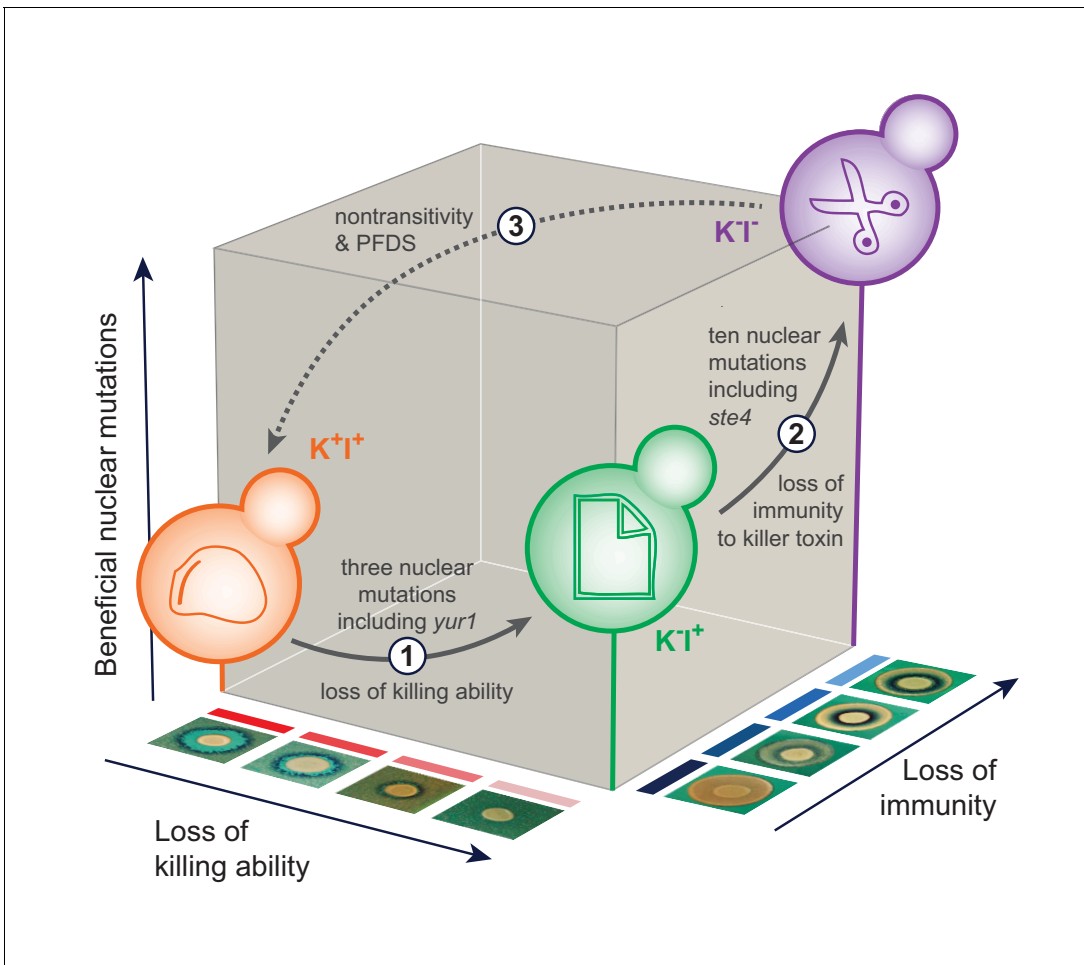

**Figure 6.** The sequence of events leading to the evolution of nontransitivity in population BYS1-D08. Nontransitivity arises through multilevel selection requiring adaptive mutations in both the nuclear and viral genomes. The Early clone (orange) produces, and is resistant to, killer toxin. Step 1: after 335 generations, the Intermediate clone (green) fixed three nuclear mutations including a beneficial mutation in *yur1* and lost the ability to produce killer toxin due to intracellular competition between viral variants. Step 2: after another 665 generations, the Late clone (purple) fixed an additional 10 nuclear mutations including a beneficial mutation in *ste4* and lost immunity to the killer toxin, which is no longer present in the environment. Step 3: when brought into competition with the Early clone (1000 generations removed), the Late clone loses in a frequency-dependent manner due to killer toxin produced by the Early clone. Positive frequency-dependent selection (PFDS) emerges in the competition because the fitness disadvantage of the Late clone can be overcome if it starts the competition at high frequency relative to the Early clone.

observe complex dynamics including genetic hitchhiking and clonal interference in the host populations as well as the intracellular viral populations. Phenotypic and genotypic changes including the loss of killing ability, mutations in the host-encoded cell-wall biosynthesis genes, and the virally encoded toxin genes occur repeatedly across replicate populations. The loss of killer-associated phenotypes—killing ability and immunity to the killer toxin—leads to three phenomena with implications for adaptive evolution: positive frequency-dependent selection, multilevel selection, and nontransitivity.

Frequency-dependent selection can be either negative, where rare genotypes are favored, or positive, where rare genotypes are disfavored. Of the two, negative frequency-dependent selection is more commonly observed in experimental evolution, arising, for example, from nutrient cross-feeding (*Helling et al., 1987*; *Turner et al., 1996*; *Spencer et al., 2008*; *Kinnersley et al., 2014*; *Plucain et al., 2014*; *Green et al., 2020*) and spatial structuring (*Rainey and Travisano, 1998*; *Frenkel et al., 2015*). Positive frequency-dependent selection, in contrast, is not typically observed in experimental evolution. By definition, a new positive frequency-dependent mutation must invade an established population at a time when its fitness is at its minimum. Even in situations in which

positive frequency-dependent selection is likely to occur, such as the evolution of cooperative group behaviors and interference competition (*Chao and Levin, 1981*), a mutation may be unfavorable at the time it arises. A crowded, structured environment provides an opportunity for allelopathies to offer a local advantage. Here we describe an alternative mechanism for the success of positive frequency-dependent mutations through multilevel selection of the host genome and a toxin-encoding intracellular virus. The likelihood of such a scenario occurring is aided by the large population size of the extrachromosomal element: each of the $\sim10^5$ cells that comprise each yeast population contains $\sim10^2$ viral particles (*Bostian et al., 1983*; *Ridley and Wickner, 1983*).

Nontransitivity in our experimental system is due, in part, to interference competition. The production of a killer toxin by the Early clone kills the toxin-susceptible Late clone in a frequency-dependent manner: higher starting frequencies of the Early clone result in higher concentrations of toxin in the environment. Interference competition can drive ecological nontransitivity (*Kerr et al., 2002*; *Kirkup and Riley, 2004*), suggesting that similar mechanisms may underlie both ecological and genealogical nontransitivity. The adaptive evolution of genealogical nontransitivity in our system does not follow the canonical model of a co-evolutionary arms race where the host evolves mechanisms to prevent the selfish replication of the virus and the virus evolves to circumvent the host's defenses (*Daugherty and Malik, 2012*; *Rowley, 2017*). Rather, mutations that fix in the viral and yeast populations do so because they provide a direct fitness advantage in their respective populations. Nontransitivity arises through the combined effect of beneficial mutations in the host genome (which improves the relative fitness within the yeast population, regardless of the presence or absence of the killer virus) and the adaptive loss of killing ability and degeneration of the intracellular virus (which provides an intracellular fitness advantage to the virus). The end result is a high-fitness yeast genotype (relative to the ancestral yeast genotype) that contains degenerate viruses, rendering their hosts susceptible to the virally encoded toxin.

Though we did not find an impact of nuclear mutations on killer-associated phenotypes, we do observe a statistical enrichment of mutations in genes involved in β-glucan biosynthesis and in genes that when deleted confer a high level of resistance to the killer toxin. Nearly all mutations in these toxin-resistance genes are nonsynonymous (18 nonsense/frameshift, 21 missense, one synonymous), indicating a strong signature of positive selection. This suggests that the nuclear genome adapts in response to the presence of the killer toxin; however, the effect of these mutations may be beyond the resolution of our fitness assay.

Among the viral variants we identified were two unique $\sim1$ kb deletions; remnants of the killer virus that retain little more than the *cis*-acting elements necessary for viral replication and packaging. These defective interfering particles are thought to outcompete full-length virus due to their decreased replication time (*Kane et al., 1979*; *Ridley and Wickner, 1983*; *Esteban and Wickner, 1988*). Defective interfering particles are common to RNA viruses (*Holland et al., 1982*). Though there are several different killer viruses in yeast (e.g. K1, K2, K28, and Klus), each arose independently and has a distinct mechanism of action (*Rodríguez-Cousiño et al., 2017*). Nontransitive interactions may therefore arise frequently through cycles of gains and losses of toxin production and toxin immunity in lineages that contain RNA viruses.

Reports of nontransitivity arising along evolutionary lines of descent are rare (*de Visser and Lenski, 2002*; *Beaumont et al., 2009*). The first (and most widely cited) report of nontransitivity along a direct line of descent occurred during yeast adaptation in glucose-limited chemostats (*Paquin and Adams, 1983*). This experiment was correctly interpreted under the assumption—generally accepted at the time—that large asexual populations evolved by clonal replacement, where new beneficial mutations arise and quickly sweep to fixation. This strong selection/weak mutation model, however, is now known to be an oversimplification for large asexual populations, where multiple beneficial mutations arise and spread simultaneously through the population leading to extensive clonal interference (*Gerrish and Lenski, 1998*; *Kvitek and Sherlock, 2013*; *Lang et al., 2013*). In addition, the duration of the Paquin and Adams experiment was too short for the number of reported selective sweeps to have occurred (4 in 245 generations and 6 in 305 generations, for haploids and diploids, respectively). The strongest known beneficial mutations in glucose-limited chemostats, hexose transporter amplifications, provide a fitness advantage of $\sim30\%$ (*Gresham et al., 2008*; *Kvitek and Sherlock, 2011*) and would require a minimum of $\sim150$ generations to fix in a population size of $4 \times 10^9$ (*Otto and Whitlock, 1997*). We contend that Paquin and Adams observed nontransitive interactions among contemporaneous lineages—ecological nontransitivity—rather than

nontransitivity among genealogical descendants. Apart from the present study, there are no other examples of nontransitivity arising along a line descent, but numerous examples of nontransitive interactions among contemporaneous lineages (*Sinervo and Lively, 1996*; *Kerr et al., 2002*; *Kirkup and Riley, 2004*; *Károlyi et al., 2005*; *Laird and Schamp, 2006*; *Reichenbach et al., 2007*; *Precoda et al., 2017*; *Menezes et al., 2019*).

Here we present a mechanistic case study on the evolution of nontransitivity along a direct line of genealogical descent. We determine the specific nuclear and viral changes that lead to nontransitivity in our focal population (*Figure 6*). Our results show that the continuous action of selection can give rise to genotypes that are less fit compared to a distant ancestor. We show that nontransitive interactions can arise quickly due to multilevel selection in a host/virus system. In the context of this experiment multi-level selection is common—most yeast populations fix nuclear and viral variants by Generation 1000. Overall, our results demonstrate that adaptive evolution is capable of giving rise to nontransitive fitness interactions along an evolutionary lineage, even under simple laboratory conditions.

# Materials and methods

## Key resources table

| Reagent type (species) or resource | Designation | Source or reference | Identifiers | Additional information |
|---|---|---|---|---|
| Strain, strain background (*Saccharomyces cerevisiae*) | yGIL432 | First reported in *Lang et al., 2011* | Early clone/ Ancestor of evolution experiment | Lang Lab strain collection |
| Strain, strain background (*Saccharomyces cerevisiae*) | yGIL519 | First reported in *Lang et al., 2011* | ymCitrine reference strain | Lang Lab strain collection |
| Strain, strain background (*Saccharomyces cerevisiae*) | yGIL1582 | This paper | Intermediate clone | Lang Lab strain collection |
| Strain, strain background (*Saccharomyces cerevisiae*) | yGIL1042 | This paper | Late clone | Lang Lab strain collection |
| Strain, strain background (*Saccharomyces cerevisiae*) | yGIL1097 | This paper | Sensitive tester strain | Lang Lab strain collection |
| Strain, strain background (*Saccharomyces cerevisiae*) | yGIL1253 | This paper | Early clone/ Ancestor (M1 cured) | Lang Lab strain collection |
| Strain, strain background (*Saccharomyces cerevisiae*) | yGIL1353 | This paper | kar1Δ15 mating partner | Lang Lab strain collection |
| Recombinant DNA reagent | pMR1593 | Mark Rose (Georgetown University) | *kar1Δ15* integrating plasmid | ATCC (87710) |
| Sequence-based reagent | M1_F1 | This paper | PCR (Sanger sequencing) | TTGGCTATTACAGCGTGCCA |
| Sequence-based reagent | M1_F5 | This paper | PCR (Sanger sequencing) | ATGACGAAGCCAACCCAAGT |
| Sequence-based reagent | M1_F7 | This paper | PCR (Sanger sequencing) | CAGAAAAAGAGAGAACAGGAC |

*Continued on next page*

*Continued*

| Reagent type (species) or resource | Designation | Source or reference | Identifiers | Additional information |
|---|---|---|---|---|
| Sequence-based reagent | M1_R3 | This paper | cDNA synthesis, PCR (Sanger sequencing) | TGCTGTTGCATTAAACCAGGC |
| Sequence-based reagent | M1_R6 | This paper | PCR (Sanger sequencing) | ATAGCCCGGTGCTCTGTAGG |
| Sequence-based reagent | LA_F2 | This paper | PCR (Sanger sequencing) | ATCAGGTGATGCAGCGTTGA |
| Sequence-based reagent | LA_F3 | This paper | PCR (Sanger sequencing) | ACTCCCCATGCTAAGATTTGTT |
| Sequence-based reagent | LA_R2 | This paper | PCR (Sanger sequencing) | CGGCACCCTTACGGAGATAC |
| Sequence-based reagent | LA_R3 | This paper | PCR (Sanger sequencing) | GACCTGTAATGCCCGGAGTG |
| Sequence-based reagent | LA_R6 | This paper | PCR (Sanger sequencing) | AGTACTGAGCCCCAAGACCA |
| Sequence-based reagent | I292M_read1 | This paper | PCR (Illumina sequencing) | CGTCGGCAGCGTCAGATGTGTATAAG AGACAGNNNNNNNNCCATGGTGTC GGCTAATGGT |
| Sequence-based reagent | I292M_read2 | This paper | PCR (Illumina sequencing) | CGTGGGCTCGGAGATGTGTATAA GAGACAGAGGTCAGACACGATGCCCTA |
| Sequence-based reagent | frameshift_read1 | This paper | PCR (Illumina sequencing) | CGTCGGCAGCGTCAGATGTG TATAAGAGACAGNNNNNNNN CCCGTCTGCGACAGTAGAAA |
| Sequence-based reagent | frameshift_read2 | This paper | PCR (Illumina sequencing) | CGTGGGCTCGGAGATGTGTAT AAGAGACAGTGTGTAAG AACTGCGTGGGT |
| Sequence-based reagent | i5_adapter | This paper | PCR (Illumina sequencing) | AATGATACGGCGACCACCGAG ATCTACACNNNNNNNNTCGT CGGCAGCGTCAGATG |
| Sequence-based reagent | i7_adapter | This paper | PCR (Illumina sequencing) | CAAGCAGAAGACGGCATACGAGAT NNNNNNNNGTCTCGTGGGCTC GGAGATGTG |

## Experimental evolution

Details of the evolution experiment have been described previously (*Lang et al., 2011*). Briefly, population BYS1-D08 is one of ~600 populations that were evolved for 1000 generations at 30°C in YPD + A and T (yeast extract, peptone, dextrose plus 100 µg/ml ampicillin and 25 µg/ml tetracycline to prevent bacterial contamination). Each day populations were diluted 1:$2^{10}$ into 128 µl of YPD + A and T in round-bottom 96-well plates using a BiomekFX liquid handler. The dilution scheme equates to 10 generations of growth per day at an effective population size of ~$10^5$.

## Growth conditions and strain construction

Unless otherwise specified, yeast strains were propagated at 30°C in YPD + A and T. The ancestor and evolved populations were described previously (*Lang et al., 2011*). Early, Intermediate, and Late clones were isolated by resurrecting population BYS1-D08 at the Generation 0, 335, and 1000, respectively. These specific time points were selected to coincide with the completion of a selective sweep (*Lang et al., 2013*), when the population is expected to be near clonal. For each time point we isolated multiple clones from a YPD plate and assayed each one to verify that the killer phenotype was uniform.

The ancestral strain was cured of the M1 and LA viruses by streaking to single colonies on YPD agar and confirmed by halo assay, PCR, and sequencing. We integrated a constitutively expressed fluorescent reporter (pACT1-ymCitrine) at the *CAN1* locus in the cured ancestral strain as well as the Intermediate (Generation 335) and Late (Generation 1000) clones.

Karyogamy mutants were constructed by introducing the *kar1Δ15* allele by two-step gene replacement in the cured a *MATα* version of the ancestor (*Georgieva and Rothstein, 2002*). The *kar1Δ15*-containing plasmid pMR1593 (Mark Rose, Georgetown University) was linearized with BglII prior to transformation and selection on -Ura. Mitotic excision of the integrated plasmid was selected for plating on 5-fluorotic acid (5-FOA). Then we replaced NatMX with KanMX to enable selection for recipients during viral transfer.

## Fitness assays

Competitive fitness assays were performed as described previously (*Lang et al., 2011*; *Lang et al., 2013*). To investigate frequency dependence, competitors were mixed at various ratios at the initiation of the experiment. Competitions were performed for 50 generations under conditions identical to the evolution experiment (*Lang et al., 2011*). Every 10 generations, competitions were diluted 1:1000 in fresh media and an aliquot was sampled by BD FACS Canto II flow cytometer. Flow cytometry data was analyzed using FlowJo 10.3. Relative fitness was calculated as the slope of the change in the natural log ratio between the experimental and reference strain. To detect frequency-dependent selection, each 10-generation interval was analyzed independently to calculate starting frequency and fitness.

## Halo assay

Killer phenotype was measured using a high-throughput version of the standard halo assay (*Crabtree et al., 2019*) and a liquid handler (Biomek FX). Assays were performed using YPD agar that had been buffered to pH 4.5 (citrate-phosphate buffer), dyed with methylene blue (0.003%), and poured into a 1-well rectangular cell culture plate.

Killing ability was assayed against a sensitive tester strain (yGIL1097) that was isolated from a separate evolution experiment initiated from the same ancestor. The sensitive tester was grown to saturation, diluted 1:10, and spread (150 µl) evenly on the buffered agar. Query strains were grown to saturation, concentrated 5×, and spotted (2 µl) on top of the absorbed lawn (*Figure 1—figure supplement 2*, left).

Immunity was assayed against the ancestral strain (yGIL432). Query strains were grown to saturation, diluted 1:32, and spotted (10 µl) on the buffered agar. The killer tester was grown to saturation, concentrated 5×, and spotted (2 µl) on top of the absorbed query strain (*Figure 1—figure supplement 2*, right).

Plates were incubated at room temperature for 2–3 days before assessment. Killer phenotype was scored according to the scale in shown in *Figure 2*.

## Viral RNA isolation, cDNA Synthesis, and PCR

Nucleic acids were isolated by phenol–chloroform extraction and precipitated in ethanol. Isolated RNA was reverse-transcribed into cDNA using ProtoScript II First Strand cDNA Synthesis Kit (NEB) with either the enclosed Random Primer Mix or the M1-specific oligo M1_R3.

## Sanger sequencing and bioinformatics analyses

PCR was performed on cDNA using Q5 High-Fidelity Polymerase (NEB). The K1 ORF was amplified using primers M1_F1 or M1_F5 and M1_R6. The M1 region downstream of the polyA stretch was amplified using M1_F7 and M1_R3. The LA virus was amplified using LA_F2 and LA_R2, LA_F2 and LA_R3, or LA_F3 and LA_R6. PCR products were Sanger sequenced by Genscript.

Mutations were identified and peak height quantified using 4Peaks (nucleobytes). For intracellular competitions, mutation frequency was quantified by both Sanger and Illumina sequencing (see below), with both methods producing nearly identical results (*Figure 5—figure supplement 2*).

The Sanger sequencing data was aligned to publicly available M1 and LA references (GenBank Accession Numbers U78817 and J04692, respectively) using ApE (A plasmid Editor). The ancestral M1 and LA viruses differed from the references at 7 sites (including 3 K1 missense mutations) and 19 sites, respectively (*Figure 3—figure supplement 1*).

## Viral transfer

Viruses were transferred to *MAT**a*** strains using the *MATα* karyogamy mutant as an intermediate. Viral donors (*MAT**a***, *ura3*, NatMX) were first transformed with the pRS426 (*URA3*, 2μ ORI) for future indication of viral transfer. Cytoduction was performed by mixing a viral donor with the karyogamy mutant recipient (*MATα*, *ura3*, KanMX) at a 5:1 ratio on solid media. After a 6 hr incubation at 30℃, the cells were plated on media containing G418 to select for cells with the recipient nuclei. Recipients that grew on -Ura (indicator of cytoplasmic mixing) and failed to grow on ClonNat (absence of donor nuclei) then served as donors for the next cytoduction. These karyogamy mutant donors (*MATα*, *URA3*, KanMX) were mixed with the selected recipient (*MAT**a***, *ura3*, NatMX) at a 5:1 ratio on solid media. After a 6 hr incubation at 30℃, the cells were plated on media containing ClonNat to select for cells with recipient nuclei. Recipients that grew on -Ura (indicator of cytoplasmic mixing) and failed to grow on G418 (absence of the donor nucleus) were then cured of the indicator plasmid by selection on 5-FOA. Killer phenotype was confirmed by halo assays and the presence of the viral variants in the recipient was verified by Sanger sequencing.

## Illumina sequencing and bioinformatics analyses

Multiplexed libraries were prepared using a two-step PCR. First, cDNA was amplified by Q5 High-Fidelity Polymerase (NEB) for 10 cycles using primers I292M_read1 and I292M_read2 or frameshift_read1 and frameshift_read2 to incorporate a random 8 bp barcode and sequencing primer binding sites. The resulting amplicons were further amplified by Q5 PCR for 15 cycles using primers i5_adapter and i7_adapter to incorporate the sequencing adaptors and indices. Libraries were sequenced on a NovaSeq 6000 (Illumina) at the Genomics Core Facility at Princeton University.

Raw FASTQ files were demultiplexed using a dual-index barcode splitter (https://bitbucket.org/princeton_genomics/barcode_splitter) and trimmed using Trimmomatic (*Bolger et al., 2014*) with default settings for paired-end reads. Mutation frequencies were determined by counting the number of reads that contain the ancestral or evolved allele (mutation flanked by five nucleotides).

## Intracellular competitions

Within-cell viral competitions were performed by propagating a heteroplasmic diploid and monitoring killer phenotype and viral variant frequency. Diploids were generated by crossing the ancestor with a cytoductant harboring either the I292M or −1 frameshift viral variant. For each viral variant, three diploid lines (each initiated by a unique mating event) were passaged every other day on buffered YPD media for a total of seven single-cell bottlenecks to minimize among-cell selection. A portion of each transferred colony was cryopreserved in 15% glycerol. Cryosamples were revived, assayed for killer phenotype, and harvested for RNA. Following RT-PCR, samples were sent for Sanger sequencing and Illumina sequencing. Variant frequency deviated from the expected frequency of 0.5 at the initial time point, presumably due to an unavoidable delay between the formation of the heteroplasmic diploid and initiation of the intracellular competition from a single colony. Alternatively, viral copy number may vary between donor and recipient cells.

## Acknowledgements

We thank Reed Wickner, Amber Rice, and members of the Lang Lab for their comments on the manuscript. This work was supported by the NIH grant 1R01GM127420 (GIL) and a Faculty Innovation Grant from Lehigh University (GIL). Illumina data of viral competitions and evolved nuclear genomes are accessible under the BioProject ID PRJNA553562 and PRJNA205542, respectively. Conceptualization and writing were performed by SWB and GIL Investigation was performed by SWB and ABR.

## Additional information

### Funding

| Funder | Grant reference number | Author |
| --- | --- | --- |
| National Institutes of Health | R01GM127420 | Gregory I Lang |
| Lehigh University | Faculty Innovation Grant | Gregory I Lang |

The funders had no role in study design, data collection and interpretation, or the decision to submit the work for publication.

### Author contributions
Sean W Buskirk, Conceptualization, Formal analysis, Investigation, Writing - original draft, Writing - review and editing; Alecia B Rokes, Formal analysis, Investigation, Writing - review and editing; Gregory I Lang, Conceptualization, Supervision, Funding acquisition, Writing - original draft, Writing - review and editing

### Author ORCIDs
Sean W Buskirk  https://orcid.org/0000-0003-1213-8130
Alecia B Rokes  http://orcid.org/0000-0001-9496-0296
Gregory I Lang  https://orcid.org/0000-0002-7931-0428

### Decision letter and Author response
Decision letter https://doi.org/10.7554/eLife.62238.sa1
Author response https://doi.org/10.7554/eLife.62238.sa2

## Additional files

### Supplementary files
• Supplementary file 1. Killer phenotype and K1 mutations in evolved yeast populations at Generation 1000. A caret (^) indicates that a population is heteroplasmic for variants listed. An asterisk (*) indicates that the mutation results in loss of PCR primer binding sites thereby preventing further characterization.

• Supplementary file 2. Mutational biases in viral and nuclear data sets.

• Transparent reporting form

### Data availability
Illumina data of viral competitions and evolved nuclear genomes are accessible under the BioProject ID PRJNA553562 and PRJNA205542, respectively.

The following dataset was generated:

| Author(s) | Year | Dataset title | Dataset URL | Database and Identifier |
|---|---|---|---|---|
| Buskirk SW, Lang GI | 2019 | Time-course sequencing of intracellular viral competitions | https://www.ncbi.nlm.nih.gov/bioproject/?term=PRJNA553562 | NCBI BioProject, PRJNA553562 |

The following previously published dataset was used:

| Author(s) | Year | Dataset title | Dataset URL | Database and Identifier |
|---|---|---|---|---|
| Lang GI, Desai MM, Botstein D | 2013 | The sequencing of *Saccharomyces cerevisiae* strains. | https://www.ncbi.nlm.nih.gov/bioproject/?term=PRJNA205542 | NCBI BioProject, PRJNA205542 |

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
