## [Decision Letter]

**Acceptance summary:**

Whilst non-transitive competitive interactions (where A beats B, B beats C, and C beats A) have been found in ecology, their occurrence within a single evolving lineage has not been shown rigorously before. Authors evolved yeast cells infected with a "Killer" virus, and found that selection occurring at multiple scales – among viruses within a cell – and between their host cells within a population – results in evolved populations that are less fit than the ancestors. This work demonstrates that evolution may not be a simple linear march of progress. Rather, progress over short time scales can sometimes lead to a reduction of fitness over the longer time scale due to the evolution of ecological interactions. For future work, we would like to know whether non-transitivity can also occur when selection acts only at one level, and how common non-transitivity is, what features predict its evolution, and at what evolutionary scale it acts.

**Decision letter after peer review:**

Thank you for submitting your article "Adaptive evolution of nontransitive fitness in yeast" for consideration by *eLife*. Your article has been reviewed by three peer reviewers, including Wenying Shou as the Reviewing Editor and Reviewer #1, and the evaluation has been overseen by Naama Barkai as the Senior Editor. The following individuals involved in review of your submission have agreed to reveal their identity: Kristina Hillesland (Reviewer #3); Duncan Greig (Reviewer #4).

The reviewers have discussed the reviews with one another and the Reviewing Editor has drafted this decision to help you prepare a revised submission.

Summary:

The findings presented in this manuscript are interesting. They show that selection is happening at multiple scales – among viruses within a cell – and between their host cells within a population. The conflict between these levels of selection results in evolved populations that are less fit than the ancestors. This work demonstrates that evolution may not be a simple linear march of progress. Rather, progress over short time scales can sometimes lead to a reduction of fitness over the longer time scale due to the evolution of ecological interactions.

The reviewers, who all identified themselves, have a range of points that they would like to see addressed, and for ease, the verbatim reviews are attached, including a summary of the discussion among the reviewers.

Reviewer #1:

Buskirk et al. examined the evolution of nontransitive fitness effects in yeast. They showed that during evolution in rich glucose medium, a late clone (1000 generations) outcompeted an intermediate clone (300 generations), but lost in direct competition with the ancestor (in a frequency-dependent fashion: late clone when rare loses to ancestor and when abundant outcompetes ancestor). This is due to adaptation in the nuclear genome and intracellular killer virus. Essentially, the ancestor expresses both killing and immunity phenotypes (K^+^I^+^), the intermediate clone expresses immunity (K^-^I^+^), and the late clone expresses neither (K^-^I^-^). This trend is observed in many evolving populations. In the absence of the killing interaction, virus does not affect host fitness. That is, when killing interactions are absent, fitness changes are due to mutations in the nuclear genome. Changes in killing and immunity phenotypes are driven by intracellular competition of viruses where viruses defective in killing and/or immunity have an advantage over functional viruses.

I find the work quite interesting, although I also find it a bit incomplete. If you have data that verify nuclear mutations that made intermediate clones more fit than ancestor and late clones more fit than intermediate clones, please include these. Experiments in yeast are very easy, and I think that giving one example for both cases will be helpful for molecular-oriented readers. For example, are those causative nuclear mutations completely independent of killer virus biology? However, upon discussing with the two other reviewers, I will not insist on this.

A schematic summary figure will be helpful.

Reviewer #3:

The findings presented in this manuscript are really exciting. They show that selection is happening at multiple scales – among viruses within a cell – and between their host cells within a population. The conflict between these levels of selection results in evolved populations that are less fit than the ancestors. This result is exciting because it happens repeatedly in independently-evolving populations, showing that it can be a general result. It is also an example of how a non-transitive interaction can evolve de novo, as the authors claim in the manuscript. The experiments seem to rule out most alternative hypotheses. However, the authors could explain their reasoning more clearly in some cases.

1) In particular I found it difficult to understand some of their conclusions in the –subsection “Host/virus co-evolutionary dynamics are complex and operate over multiple scales”, without rereading, rewriting results, and lots of thinking. They state that production of active toxin or maintenance of the virus has no detectable fitness cost to the host. There are a lot of comparisons to think through here to get to that conclusion, and I think the average reader needs to be taken through that. Even though I have some experience thinking about costs and how they can be estimated, I still spent quite a lot of time trying to follow the logic from Figure 5A to that statement. In fact, I still do not understand how they are distinguishing between “production of active toxin” and “maintenance of the virus”. I also had to spend a lot of time thinking through the results in Figure 3 and the conclusion stated in the aforementioned subsection”.

2) I think it would be helpful to the reader, and interesting, if there were more of an explanation about WHY K^+^I^+^ cells have positive frequency-dependent fitness relative to K^-^I^-^ cells. Why is the presence of an active virus and immunity more beneficial at higher frequencies?

Reviewer #4:

This study shows how well mixed populations of yeast cells initially expressing both an anticompetitor toxin and resistance to it, first lose toxin production (because there is a cost but no benefit to toxin production when all cells are resistant) and then lose resistance (because there is a cost but no benefit to resistance when no cells produce toxin). Consequently, these evolved sensitive populations have lower fitness than their own toxin-producing (resurrected) ancestors, but only if the toxic ancestors are introduced at a high enough frequency, that is, there is positive frequency dependent selection. These results are quite intuitive and satisfying, and are well supported by rigorous experiments determining the causal mutations and their selective advantages both within intra-cellular populations of the virus, and between cells in a the evolving populations. This was really nice, thorough, and interesting work. However the overall result is not really surprising, as much similar work has been done before (and is properly cited) in which three types of competitors show non-transitive pairwise fitness relationships.

The main claim to originality is that the three types here are generated sequentially by two rounds of mutation, natural selection, and replacement/fixation: that is, there is genealogical nontransitivity between ancestors and descendants, rather than just ecological nontransitivity between contemporary co-existing variants. This demonstrates an important principle: that natural selection can produce a decline in overall relative fitness in a lineage over multiple rounds of mutation and fixation. The only other reported example of this in experimental evolution is the work of Paquin and Adams, 1983, but the authors here argue convincingly that the Paquin and Adams, lacking the benefit of sequencing to identify mutations and their frequencies, inadvertently competed ecological types that were co-existing in their evolving populations and had not fixed.

My only criticism, then, is that the example of non-transitivity demonstrated here is rather "obvious"; the result is entirely predictable, given the amount of previous work in similar microbial systems. However, this is countered by the fundamental nature of the question for evolutionary biology, and the lack of specific experimental examples, apart from the very old Paquin and Adams. Overall, then, I am satisfied that this paper is a significant step forward, and is appropriate for publication in *eLife*. I found it well written, interesting, and the conclusions were well supported by careful and thorough experiments.

---

## [Author Response]

Reviewer #1:Buskirk et al. examined the evolution of nontransitive fitness effects in yeast. They showed that during evolution in rich glucose medium, a late clone (1000 generations) outcompeted an intermediate clone (300 generations), but lost in direct competition with the ancestor (in a frequency-dependent fashion: late clone when rare loses to ancestor and when abundant outcompetes ancestor). This is due to adaptation in the nuclear genome and intracellular killer virus. Essentially, the ancestor expresses both killing and immunity phenotypes (K^+^I^+^), the intermediate clone expresses immunity (K^-^I^+^), and the late clone expresses neither (K^-^I^-^). This trend is observed in many evolving populations. In the absence of the killing interaction, virus does not affect host fitness. That is, when killing interactions are absent, fitness changes are due to mutations in the nuclear genome. Changes in killing and immunity phenotypes are driven by intracellular competition of viruses where viruses defective in killing and/or immunity have an advantage over functional viruses.I find the work quite interesting, although I also find it a bit incomplete. If you have data that verify nuclear mutations that made intermediate clones more fit than ancestor and late clones more fit than intermediate clones, please include these.

Yes, we do have data on the fitness effects of the mutations that differ between the Early, Intermediate, and Late clones. Though it was measured out of the context of the evolution experiment, we previously identified *yur1* and *ste4* as beneficial mutations (Buskirk et al. 2017). We now indicate this in the manuscript (Results).

Experiments in yeast are very easy, and I think that giving one example for both cases will be helpful for molecular-oriented readers. For example, are those causative nuclear mutations completely independent of killer virus biology? However, upon discussing with the two other reviewers, I will not insist on this.A schematic summary figure will be helpful.

This is a great suggestion, and it prompted us to add an additional figure (Figure 6), which provides a concise illustration of the of events leading to the evolution of nontransitivity and serves as a nice summary of the manuscript. We reference Figure 6 in the Discussion.

Reviewer #3:The findings presented in this manuscript are really exciting. They show that selection is happening at multiple scales – among viruses within a cell – and between their host cells within a population. The conflict between these levels of selection results in evolved populations that are less fit than the ancestors. This result is exciting because it happens repeatedly in independently-evolving populations, showing that it can be a general result. It is also an example of how a non-transitive interaction can evolve de novo, as the authors claim in the manuscript. The experiments seem to rule out most alternative hypotheses. However, the authors could explain their reasoning more clearly in some cases.1) In particular I found it difficult to understand some of their conclusions in the –subsection “Host/virus co-evolutionary dynamics are complex and operate over multiple scales”, without rereading, rewriting results, and lots of thinking. They state that production of active toxin or maintenance of the virus has no detectable fitness cost to the host. There are a lot of comparisons to think through here to get to that conclusion, and I think the average reader needs to be taken through that.Even though I have some experience thinking about costs and how they can be estimated, I still spent quite a lot of time trying to follow the logic from Figure 5A to that statement. In fact, I still do not understand how they are distinguishing between “production of active toxin” and “maintenance of the virus”. I also had to spend a lot of time thinking through the results in Figure 3 and the conclusion stated in the aforementioned subsection.

We edited these sections to more carefully walk the reader through our data.

2) I think it would be helpful to the reader, and interesting, if there were more of an explanation about WHY K^+^I^+^ cells have positive frequency-dependent fitness relative to K^-^I^-^ cells. Why is the presence of an active virus and immunity more beneficial at higher frequencies?

We have added a sentence in the Discussion to clarify this point: “The production of a killer toxin by the Early clone kills the toxin-susceptible Late clone in a frequency-dependent manner: higher starting frequencies of the Early clone result in higher concentrations of toxin in the environment.” We also added the phrase “due to a high local concentration of secreted toxin” to the end of the sentence “Killer toxin has been shown to impart frequency-dependent selection in structured environments” in the Results.